# Modelling the mass adoption potential of wearable medical devices

Qing Yang[1], Abdullah Al Mamun[1]*, Naeem Hayat[2], Mohd Fairuz Md. Salleh[1], Gao Jingzu[3], Noor Raihani Zainol[4]

**1** UKM - Graduate School of Business, Universiti Kebangsaan Malaysia, UKM Bangi, Bangi, Malaysia, **2** Global Entrepreneurship Research and Innovation Centre, Universiti Malaysia Kelantan, Kota Bharu, Malaysia, **3** UCSI Graduate Business School, UCSI University, Cheras, Kuala Lumpur, Malaysia, **4** Faculty of Entrepreneurship and Business, Universiti Malaysia Kelantan, Kota Bharu, Malaysia

* almamun@ukm.edu.my, mamun7793@gmail.com

**Data Availability Statement:** All relevant data are within the paper and its Supporting information files.

**Funding:** The author(s) received no specific funding for this work.

## Abstract

Digital technologies empower users to manage their health and reduce the burden on the public health system. The mass adoption of wearable medical devices (WMDs) promotes the ageing population's confidence besides facilitating users. Thus, the current study aims to empirically evaluate the formation of perceived product value (PPV) with the WMDs' computability, usefulness, cost, and accuracy, the intention to use WMDs influenced by health consciousness (HCS), health anxiety (HAY), product value, and perceived critical mass (PCM), and later the adoption of WMDs among Chinese adults. The study examined the mediating effect of PPV on the relationship between the intention to use WMDs and perceived compatibility (PCT), perceived cost (PCO), perceived usefulness (PUS), and perceived technology accuracy (PTA). This study adopted a cross-sectional approach and used an online survey to collect quantitative data from 1,160 Chinese adults. Data analysis was performed using the partial least squares structural equation modelling (PLS-SEM). Results showed that PCT, PUS, and PTA significant positive effect on PPV. Meanwhile, HCS, PCM, and PPV has a significant positive effect on intention to use WMDs, and the intention to use WMDs and PCM influenced the adoption of WMDs. Consequently, the analysis confirmed that PPV mediated the relationships between the intention to use WMD and PCT, PUS, and PTA. The WMD cost must be reduced to enhance the value of WMDs. Finally, the study's implications, limitations, and suggestions for future studies are discussed.

## 1. Introduction

With the recent advancements in digital technology and the COVID-19 pandemic crisis, new healthcare innovations, such as wearable medical devices (WMDs), have generated a surge of enthusiasm among healthcare technology users [1]. Users are looking for technology that can facilitate and meet their demands to instantly and precisely depict their health condition [2]. On the other hand, the ageing population has been increasing, whereby about 30% of the

**Competing interests:** The authors have declared that no competing interests exist.

world population is in its early 50s. Users from this group are looking for individualised healthcare services that include self-management or health assessment devices [3]. The increased demand for healthcare technology instigates the infiltration of different types of medical devices and smart healthcare wearable devices (WDs) that can monitor and track fitness [4].

The market for WDs is constantly growing, and it could reach USD 30 billion by early 2023 [3]. Technology firms are attracted to the growing market of WDs. New applications and features are added to WDs to enable users to manage their health at their convenience [5]. Furthermore, WDs allow the users to screen their daily health conditions such as physical movement, quality of food intake, diet nutrition value, sleep quality, pulse rate, and general health state [6].

Numerous types of WDs are available in the market and these devices enable users to monitor and manage their health with technologies that come in the shape of smartwatches, smart bands, and mobile phone-based gadgets [2]. For example, Chinese consumers have brands such as Fitbit, Apple Watch, Samsung Galaxy Watch, and Honor Smart Watch to choose from [1]. These devices allow users to screen and manage health conditions at their convenience [6]. Currently available smartphone-based WMDs include various types of motion sensors and bio-sensors that record feedback on a patient's mobility status and other physical activities based on data collected daily, allowing for a more objective assessment of possible therapies [7,8]. Such WMD data can be utilised alone or in combination with pharmaceuticals, devices, or other treatments to improve patient care and health outcomes, especially in the case of chronic conditions [9]. Thus, WMDs help users to reduce health risks and screen their body temperature, heartbeat rate, daily mobility, and other health conditions [10].

China is the most inhabited nation in the world, and 18% of the country's population is aged more than 50 years old [11]. The public requires promising health facilities as public health services may not be available to everyone when they need them. Moreover, healthcare technology can facilitate the large population to gain control of personal care and support in building efficacy in the public healthcare system [12]. Taking care of personal health reduces the burden on the public healthcare system and empowers the general public to maintain their health [12].

However, the question remains as to how the WMD technological attributes, personal health behaviour, and social mass adoption instigate the intention to use and the adoption of WMDs. Therefore, the current study explores the construction of WMD value with the technology features (compatibility, cost, usefulness, and accuracy) forming the product value. Besides, this study examines the effects of health consciousness, health anxiety, perceived product value, and perceived critical mass on the intention to use and the adoption of WMDs.

## 2. Literature review

### 2.1 Theoretical foundation

Technological attributes significantly influence the value of the technology and its adoption later. The technological aspects of compatibility, usefulness, cost, and accuracy offer the perception of value and build the necessary conditions for using healthcare technology [13]. The technology adoption model (TAM) considers the technological aspects that can lead to the intention to use and adoption of technology. Nonetheless, the individual personal behaviour and technological attributes are essential for forming the intention to use the technology [14]. In the case of healthcare technologies, HCS and HAY promote the use of healthcare technologies that can facilitate the management of personal health at the users' convenience [15].

Moreover, the social adoption of technology promotes the intention to use WMDs [16]. The perception of acceptance and consideration that the technology is useful and benefits the users grows as a result of the widespread adoption among the community and peers [17]. This current study's model extends the TAM with users' health behaviour and social aspect of mass adoption that promotes the intention to use and adoption of the healthcare technology [18]. The social acceptance of healthcare technology also benefits public healthcare services [19]. The formation of the intention to use and adoption of healthcare technology is the behavioural process that emerges from the technological, personal, and social factors, accurately offering the desired features to the consumers.

## 2.2 Hypotheses development

**2.2.1 Perceived compatibility (PCT).** PCT refers to how well a new technology integrates with existing technologies without significantly affecting their functionality [15]. For healthcare technology, compatibility is described as the ability to transfer health-related data to mobile devices and increase the users' well-being, which then influences patients' willingness to continue using [20]. A higher degree of compatibility between new and old technologies is positively related to users' intention to use them in the future [21]. Besides, compatibility is described as aligning innovation with existing product values, current needs, and lifestyle of potential consumers [22]. Wang et al. [4] have postulated that technology compatibility builds the value of the technology products. PCT is regarded as a critical aspect in the adoption of new technologies and considerably impacts users' behavioural intentions [12,15]. Thus, the following hypothesis is proposed:

*H1a*: *PCT has a positive effect on the PPV of WMDs.*

**2.2.2 Perceived cost (PCO).** A critical aspect that determines customer acceptance of technology is PCO [23]. The degree to which a person believes that using WMDs would cost money is known as PCO [24]. The higher the PCO of WMDs, the less likely they will be used. Users generally look for high-quality products at a reasonable and lower cost [25]. If customers are to use technological innovations, the devices must be reasonably priced compared to alternatives; otherwise, user acceptance of the new technology may not be practical [23]. Hence, this study proposes the following hypothesis:

*H1b*: *PCO has a positive effect on the PPV of WMDs.*

**2.2.3 Perceived usefulness (PUS).** Davis [18] defines PUS as 'the degree to which a person believes that utilising a certain system will improve his or her performance'. In the current study's context, PUS is defined as how individuals believe that using WMDs would improve their health status [15,20]. When a person perceives a medical device as a useful technological tool, his or her intention to use it will lead to the adoption of the new technology [1]. Users are more likely to adopt WMDs when they believe the devices would improve their lives [12,26]. Furthermore, the degree to which Chinese adults consider WMDs to be simple to use would impact both their PPV and their intention to adopt the WMDs. Previous studies have shown that PUS is one of the most vital indicators of wearable technology adoption [15,22,27]. As such, based on past literature, the following hypothesis is formulated:

*H1c*: *PUS has a positive effect on the PPV of WMDs.*

**2.2.4 Perceived technology accuracy (PTA).** In various industries such as the computer, digital, and healthcare equipment's, accuracy has been extensively studied [28]. The legitimacy, precision, and reliability with which information is given are referred to as technology accuracy [29]. WDs are new healthcare technologies that can assist patients in the early detection of severe health issues [30] besides providing early assistance and alerts to general users [1]. One of the significant implementation challenges in the healthcare market is technology accuracy [12,23]. Promoting awareness and accuracy of the product value is thus critical for customers and users. Therefore, the following hypothesis is formulated:

*H1d*: *PTA has a positive effect on the PPV of WMDs.*

**2.2.5 Perceived product value (PPV).** Providing healthcare consumers with the best possible product value is undeniably crucial [31]. The perceived value of WDs is described as the overall view of wearable technologies based on their benefits and costs that attract consumers to the technology products [15]. According to Nilson [32], PPV is 'a comparison of tangible and intangible benefits from a product's generic and supplementary levels, as well as the total costs of manufacturing and usage'. The perception of product value emerges from the perception that the benefits a client receives from a product outweigh the long-term costs he or she may incur [33]. In contrast, 'behavioural intention' refers to a user's apparent desire to adopt new technology [30,34]. PPV is considered one of the most vital factors in behavioural intention, and buyers and producers have gradually recognised its great importance [31]. According to numerous information technology studies, the perceived value of using mobile internet services on portable devices has a beneficial impact on adoption intention [15,31]. As such, this study proposes the following hypothesis:

*H2a*: *PPV has a positive effect on the intention to use WMDs.*

**2.2.6 Health consciousness (HCS).** HCS refers to the degree to which health concerns are incorporated into a person's everyday activities [30]. There is a vital link between HCS and healthy behaviours. People with health cognizance have a greater grasp of their health, pay attention to health issues, and take precautions to protect their health [35]. Additionally, people who are mindful of their health will be more interested in having the correct information to monitor their health [6]. They will continuously monitor the health indicators and use the healthcare services of the product [13]. Wearable medical technology aims to change people's health behaviours and improve their health [5]. Sergueeva et al. [3] have stated that HCS is one of the most crucial factors in predicting health-related preventive behaviour. Thus, when an individual's HCS is stronger, his or her perception of and intention to use WMDs will increase. This present study proposes the following hypothesis:

*H2b*: *HCS has a positive effect on the intention to use WMDs.*

**2.2.7 Health anxiety (HAY).** HAY is when an individual has an extreme preoccupation with researching his or her health situation and the conviction that he or she is suffering from or will suffer from a serious illness that is yet to be detected [36]. Moreover, HAY is characterised as a person's fear or uneasiness due to bodily symptoms that indicate a severe illness [37]. HAY generally contributes to safety-seeking behaviour, and the goal is to protect and take control of one's health [38]. Besides, HAY causes users to engage in safety behaviours, such as wearing medical devices [6]. Anxious customers are more likely to purchase healthcare technologies to help them accomplish their goals [36]. Furthermore, individuals with a high level

of HAY use more healthcare technology and have a higher intention to use WMDs. HAY is linked to health information [37] but only a few studies have examined how HAY influences the intention to use WMDs. Therefore, the following hypothesis is formulated:

*H2c*: *HAY has a positive effect on the intention to use WMDs.*

**2.2.8 Perceived critical mass (PCM).** PCM is built on the notion that a significant percentage of a population is already using the technology [39] and is regarded as an essential feature of social influence that impacts people's behavioural intentions and adoption of new technology [16]. Ku et al. [40] describe PCM as 'the degree to which a user of a product/services believes that the people he or she communicates with are using the same product or services, and they tend to use the product/services in the future continually'. PCM indicates the point at which a large enough number of people have utilised and accepted the technology. In healthcare technology, PCM refers to a user's perception that most of his or her peers are using a WMD [39]. In the current study, PCM is defined as a WMD user's belief that many individuals with whom he or she interacts are using the device. When users believe that the use of a WMD has reached critical mass or mass adoption, they will trust and expect their friends and family, with whom they interact, to continue using it in the future, resulting in a higher intention to use the WMD [12,20]. Reaching critical mass might offer users the impression that the technology is widely accepted, thus, giving them the confidence to adopt it [14]. As such, potential users are more inclined to adopt a WMD if they believe it has reached the masses. Users will have more confidence in the WMD's long-term viability and will be more willing to adopt it [39]. Besides, empirical studies have found that the perception of critical mass directly impacts users' intent to continue using social networking services [14,39]. Hence, the following is proposed:

*H2d*: *PCM has a positive effect on the intention to use WMDs.*

**2.2.9 Adoption of WMDs.** PCM facilitates technology adoption as prospective users have enough information and the opinions of existing users. The mass adoption of technology motivates and reduces the perception of risk among new users [39]. For healthcare technologies, the mass adoption among the community enhances the adoption behaviour among new users [16]. Mass adoption among peers and family leads to confidence and trust that instigate the adoption behaviour. As such, the following hypothesis is proposed:

*H3*: *PCM has a positive effect on the adoption of WMDs.*

The intention to use health-related technology is a crucial predictor of actual adoption. According to Alam et al. [13], intention is the best predictor of adopting health-based WDs, mobile devices, or allied technologies. For example, older people require immediate personal health attention and their intention to use health-based personal devices predicts WMD adoption [23]. When a consumer's behavioural intention is stronger, he or she is more inclined to accept new technologies. Thus, this study proposes the following hypothesis:

*H4*: *Intention to use WMDs has a positive effect on the adoption of WMDs.*

**2.2.10 Mediating effect of PPV.** PPV is a multidimensional notion that can be influenced by various factors such as PCT, PCO, PUS, and PTA [22]. In the current study, PPV acts as a mediator in the relationships between intention to use WMDs and PCT, PCO, PUS, and PTA [26]. Users of WMDs should have more opinions about the devices' effectiveness,

compatibility, usefulness, accuracy, and perceived ease of use [19]. Furthermore, the PPV must be at the core of marketers' efforts to influence consumer behaviour intention [31]. Thus, the following are proposed:

*HM1*: *The relationship between PUS and intention to use WMDs is mediated by PPV.*

*HM2*: *The relationship between PCT and intention to use WMDs is mediated by PPV.*

*HM3*: *The relationship between PCO and intention to use WMDs is mediated by PPV.*

*HM4*: *The relationship between PTA and intention to use WMDs is mediated by PPV.*

### 2.3 Theoretical framework

The independent variables (PCT, PCO, PUS, PTA, PPV, HCS, HAY, and PCM), the mediating variable (PPV), and the intention and adoption of WMDs are depicted in the study's research framework (Fig 1).

## 3. Research methodology

The ethics committee of Universiti Malaysia Kelantan decided that no formal ethics approval was required for this study because this research did not collect any medical information, had no intention to publish personal information, did not collect data from underaged respondents, and there was no known risk involved. This study was conducted in accordance with the Declaration of Helsinki. Written informed consent for participation was obtained from the survey respondents. These respondents, who answered the survey via a Google form, were requested to read the ethical statement posted at the top of the form (*There is no compensation for responding, nor is there any known risk. In order to ensure that all information will remain confidential, please do not include your name. Participation is strictly voluntary and you may refuse to participate at any time.*) and proceed only if they agree. No data were collected from anyone under the age of 18 years.

### 3.1 Sample size calculation and data collection

The sample size was calculated using G-Power 3.1 with a power of 0.95 and an effect size of 0.15. The required sample size for the model was 166 with nine predictors [41]. Meanwhile,

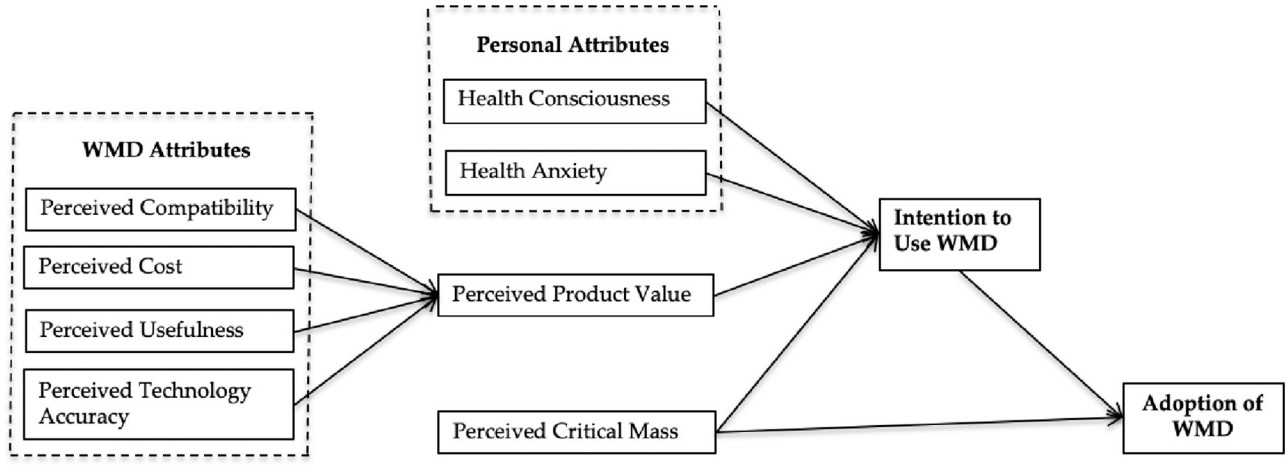

**Fig 1. Research framework.**

the partial least squares structural equation modelling (PLS-SEM) needed a minimum threshold of 200 samples [42]. This study employed convenience sampling, a non-probability method of gathering data. It is used to gather data from people who are close by and easily accessible and to select respondents who are available to the researcher [43]. In this study, data were collected using the WJX online survey form. Responses were gathered from senior citizens in China who have used WDs to monitor their health. The final data analysis was performed with 1,160 valid responses. The raw data supporting the conclusions of this study are presented as supporting material S1 Data—WMD.

## 3.2 Measurement scale

This study's measurement scale was derived from well-known and accepted scales. The items used to measure the variables and sources are reported in Appendix 1. The questionnaire was written in the English language and then translated into the Chinese language so that the respondents could understand better and reply to the questions [43]. All the questionnaire items relating to exogenous variables were marked based on a five-point Likert scale, whereas endogenous variables were graded based on a seven-point Likert scale. In the research design stage, using distinct Likert scales for input and outcome variables solves the issue of common method variance (CMV) [44].

## 3.3 Common Method Variance (CMV)

The effect of CMV as a diagnostic approach was determined using Harman's one-factor test. The single factor accounted for 36.53%, i.e. less than the threshold limit of 50.00% in Harman's one-factor test, indicating that CMV had a slight impact on this study [44]. Additionally, this study assessed CMV by testing the entire collinearity of all the constructs, as recommended by Kock [45]. Next, the variance inflation factor (VIF) for HCS (1.618), HAY (1.569), PCT (1.848), PCO (1.358), PUS (1.691), PTA (1.706), PPV (1.818), PCM (1.710), and intention to use WMDs (1.509) were all less than 3.3, demonstrating the absence of bias from the single-sourced data [44].

## 3.4 Multivariate normality

The multivariate normality for this study's data was assessed using the Web Power online tool (source: https://webpower.psychstat.org/wiki/tools/index). The calculated Mardia's multivariate skewness and kurtosis coefficient and $p$-values showed that the data had a non-normality issue since the $p$-values were below 0.05 [46].

## 3.5 Data analysis method

Using SmartPLS 3.2, this study employed the PLS-SEM technique to test the suggested model and analyse the hypotheses. The PLS-SEM technique for hypothesis testing has been validated in numerous studies and is frequently utilised [47]. Flexibility in data allocation is a feature of this technique that suits a small sample size [45]. Before examining the structural model, it is necessary to verify the constructs' reliability, convergent, and discriminant validity [42]. Cronbach's alpha measured the reliability, while Dillon-Goldstein's rho, composite reliability, and the average variance extracted (AVE) measured the internal consistency reliability [47]. In addition, the Fornell-Larcker criterion, the heterotrait-monotrait ratio (HTMT), and loadings and cross-loadings were used to assess the discriminant validity. Besides, path coefficients were used [45]. The coefficients (Beta), confidence interval, t-value, and $p$-value were used to test the hypotheses [47].

# 4. Findings

## 4.1. Demographic profile of respondents

As shown in Table 1, 52.1% of the respondents were females while 47.9% were males. The respondents' age ranges were 20–30 years (6.8%), 31–40 years (8.9%), 41–50 years (47.4%), 51–60 years (30.4%), and above 60 years old (6.5%). In terms of respondents' education level, most had a bachelor's degree (36.9%), followed by diploma (23.4%), secondary school certificate (17.0%), master's degree (16.6%), and doctoral degree (6.0%). Regarding respondents' average monthly income, 13.7% earned below CNY 2,500, 24.1% earned CNY 2,501–5,000, 25.3% earned CNY 5,001–7,500, 19.1% earned CNY 7,501–10,000, 9% earned more than CNY 12,500, and 8.8% of the respondents earned CNY 10,001–12,500. On the other hand, 24.1% have been using a medical device for more than half a year, 17.2% for more than one year, 7.8% for more than three years, 6.7% for more than five years, whereas 23.4% have never used a medical device. Finally, the respondents lived in Shanghai (13.5%), Guangdong (9.7%), Zhejiang (9.1%), Shandong (8.6%), Jiangsu (8.6%), Beijing (8.5%), Guangxi (7.8%), Hunan (5.5%), and others (28.6%).

## 4.2 PLS-SEM analysis and results

### 4.2.1 Reliability and validity.
Based on the measurement model results in Table 2, the alpha value of each construct exceeded the 0.60 benchmark [47]. Cronbach's alpha values of 0.60 to 0.70 are usually regarded as acceptable, and values higher than 0.70 are considered an excellent level of reliability [48]. The Dijkstra-Henseler's rho values for all the items were

**Table 1. Demographic characteristics.**

| | N | % | | N | % |
|---|---|---|---|---|---|
| *Gender* | | | *Education* | | |
| Male | 556 | 47.9 | Secondary school certificate | 197 | 17.0 |
| Female | 604 | 52.1 | Diploma | 272 | 23.4 |
| Total | 1160 | 100 | Bachelor's degree or equivalent | 428 | 36.9 |
| | | | Master's degree | 193 | 16.6 |
| *Age* | | | Doctoral degree | 70 | 6.0 |
| 20–30 years | 79 | 6.8 | Total | 1160 | 100 |
| 31–40 years | 103 | 8.9 | | | |
| 41–50 years | 550 | 47.4 | *Average Monthly Income* | | |
| 51–60 years | 353 | 30.4 | Below CNY 2500 | 159 | 13.7 |
| Above 60 years | 75 | 6.5 | CNY 2501- CNY 5000 | 280 | 24.1 |
| *Total* | *1160* | *100* | CNY 5001- CNY 7500 | 294 | 25.3 |
| | | | CNY 7501- CNY 10,000 | 221 | 19.1 |
| *Living Province* | | | CNY 10,001- CNY 12,500 | 102 | 8.8 |
| Beijing | 99 | 8.5 | More than CNY 12,500 | 104 | 9.0 |
| Shanghai | 157 | 13.5 | Total | 1160 | 100 |
| Guangdong | 113 | 9.7 | | | |
| Guangxi | 90 | 7.8 | | | |
| Zhejiang | 105 | 9.1 | | | |
| Shandong | 100 | 8.6 | | | |
| Hunan | 64 | 5.5 | | | |
| Jiangsu | 100 | 8.6 | | | |
| Others | 332 | 28.6 | | | |
| Total | 1160 | 100 | | | |

**Table 2. Reliability and validity.**

| Variables | No. of Items | Cronbach's Alpha | Dijkstra-Hensele's *rho* | Composite Reliability | Average Variance Extracted | Variance Inflation Factor |
|---|---|---|---|---|---|---|
| Health consciousness | 5 | 0.786 | 0.789 | 0.854 | 0.539 | 1.618 |
| Health anxiety | 5 | 0.826 | 0.826 | 0.878 | 0.590 | 1.569 |
| Perceived compatibility | 4 | 0.755 | 0.755 | 0.845 | 0.576 | 1.848 |
| Perceived cost | 3 | 0.741 | 0.741 | 0.885 | 0.794 | 1.358 |
| Perceived usefulness | 3 | 0.664 | 0.668 | 0.818 | 0.601 | 1.691 |
| Perceived technology accuracy | 5 | 0.751 | 0.752 | 0.833 | 0.500 | 1.706 |
| Perceived product value | 4 | 0.753 | 0.756 | 0.845 | 0.579 | 1.818 |
| Perceived critical mass | 4 | 0.718 | 0.719 | 0.825 | 0.541 | 1.710 |
| Intention to use WMD | 5 | 0.818 | 0.817 | 0.873 | 0.579 | 1.509 |
| Adoption of WMD | 1 | 1.000 | 1.000 | 1.000 | 1.000 | - |

**Source**: Author's data analysis.

above 0.70 except for PUS, which was 0.668. Meanwhile, the composite reliability (CR) values for all the items were above 0.70, thus, confirming the constructs' reliability and regarded as satisfactory. Next, the AVE for all the items exceeded 0.50, hence, confirming good convergent validity [47]. Lastly, the VIF must be less than 3.3 [42] and there was no issue of multicollinearity because the VIF values of all the variables were less than 3.3 [48].

The findings revealed that discriminant validity was achieved among the constructs, which were different from one another. All the variables met the Fornell-Larcker criterion because the square root of the AVE for each construct was higher than the maximum squared correlation of the variables with one another (see Appendix 2). Finally, all the values met the HTMT requirement because they were less than 0.90, indicating discriminant validity for the research constructs [42]. Loading was greater than 0.7 [48], and all the indicators of cross-loadings were greater than the values associated with the other constructs. This indicated that the indicators were correctly assigned to the corresponding three constructs and that the criteria for discriminant validity at the item level of the model were met.

**4.2.2 Study path testing.** The adjusted $r^2$ value for PPV from the four exogenous constructs (i.e., PCT, PCO, PUS, and PTA) signified that 59.8% of the variation of PPV was explained by PCT, PCO, PUS, and PTA. Meanwhile, this part of the model's $Q^2$ value was 0.341, showing medium predictive relevance [47].

Next, the path value between PCT and PPV ($\beta = 0.311$, $p = 0.000$) revealed that PCT had a positive and significant effect on PPV, thus, supporting H1a. The $f^2$ value of 0.131 indicated the small effect of PCT on PPV. On the other hand, the path value between PCO and PPV ($\beta = 0.025$, $p = 0.164$) showed that PCO had a positive but insignificant effect on PPV, hence, rejecting H1a. PUS ($\beta = 0.259$, $p = 0.000$) and PTA ($\beta = 0.336$, $p = 0.000$) exerted a positive and significant effect on PPV, thus, supporting H1c and H1d. The $f^2$ values of 0.098 and 0.167 indicated the small and medium effects of PUS and PTA on PPV, respectively. The findings are tabulated in Table 3.

The adjusted $r^2$ value for IWM from the four exogenous constructs (i.e., PPV, HCS, HAY, and PCM) showed that 54.1% of the variation of IWM was explained by PPV, PCM, HCS, and HAY. The $Q^2$ value for this part of the model was 0.310, showing medium predictive relevance [47]. The path coefficient of the relationship between PPV and IWM ($\beta = 0.495$, $p = 0.000$) revealed that PPV had a positive and significant effect on IWM, thus, supporting H2a. Besides,

**Table 3. Path coefficients.**

| No. | Path | Coefficients | CI–Min | CI–Max | t | P | $r^2$ | $Q^2$ | $f^2$ | Decision |
|---|---|---|---|---|---|---|---|---|---|---|
| Factors affecting perceived product value of WMDs | | | | | | | | | | |
| H$_{1a}$ | PCT → PPV | 0.311 | 0.261 | 0.361 | 10.328 | 0.000 | 0.598 | 0.341 | 0.131 | Accepted |
| H$_{1b}$ | PCO → PPV | 0.025 | -0.016 | 0.067 | 0.979 | 0.164 | | | 0.001 | Rejected |
| H$_{1c}$ | PUS → PPV | 0.259 | 0.210 | 0.308 | 8.494 | 0.000 | | | 0.098 | Accepted |
| H$_{1d}$ | PTA → PPV | 0.336 | 0.286 | 0.388 | 10.900 | 0.000 | | | 0.167 | Accepted |
| Intention to adopt WMDs | | | | | | | | | | |
| H$_{2a}$ | PPV → IWM | 0.495 | 0.441 | 0.546 | 15.508 | 0.000 | 0.541 | 0.310 | 0.294 | Accepted |
| H$_{2b}$ | HCS → IWM | 0.097 | 0.048 | 0.147 | 3.231 | 0.001 | | | 0.013 | Accepted |
| H$_{2c}$ | HAY → IWM | 0.033 | -0.015 | 0.082 | 1.095 | 0.137 | | | 0.001 | Rejected |
| H$_{2d}$ | PCM → IWM | 0.233 | 0.185 | 0.283 | 7.900 | 0.000 | | | 0.069 | Accepted |
| Adoption of WMDs | | | | | | | | | | |
| H$_3$ | PCM → AWM | 0.219 | 0.168 | 0.271 | 6.951 | 0.000 | 0.395 | 0.391 | 0.053 | Accepted |
| H$_4$ | IWM → AWM | 0.476 | 0.421 | 0.530 | 14.228 | 0.000 | | | 0.248 | Accepted |

**Note**: HCS: Health consciousness; HAY: Health anxiety; PCT: Perceived compatibility; PCO: Perceived cost; PUS: Perceived usefulness; PTA: Perceived technology accuracy; PPV: Perceived product value; PCM: Perceived critical mass; IWM: Intention to Use WMD; AWM: adoption of WMD.

**Source**: Author's data analysis.

the $f^2$ value of 0.294 indicated the medium effect of PPV on IWM. Meanwhile, the path value between HCS and IWM (β = 0.097, p = 0.001) showed that HCS exerted a positive and significant effect on IWM, supporting H2b. Nevertheless, the path value for HAY (β = 0.033, p = 0.137) displayed a positive but insignificant effect on IWM, hence, rejecting H2c. Finally, the path value for PCM (β = 0.233, p = 0.000) revealed that PCM exerted a positive and significant effect on IWM, thus, supporting H2d. The $f^2$ value of 0.069 indicated the small effect of PCM on IWM. Table 3 presents the results.

The adjusted $r^2$ value for AWM with the two input constructs (i.e., PCM and IWM) demonstrated that 39.5% of the variation of AWM was elucidated by PCM and IWM. The $Q^2$ value of the model was 0.391, showing high predictive relevance [47]. The effect of PCM (β = 0.219, p = 0.000) and IWM (β = 0.476, p = 0.000) was positive and significant on AWM, thus, supporting H3 and H4. The $f^2$ values of 0.053 and 0.248 indicated the small and medium effects of PCM and IWM on AWM, respectively. All the findings are listed in Table 3.

**4.2.3 Mediational analysis.** The mediation analysis showed that PPV mediated the relationship between PCT and IWM (β = 0.154, p = 0.000), thus, supporting HM1. Next, the path value for PCO (β = 0.012, p = 0.164) indicated that PPV did not mediate the relationship between PCO and IWM, hence, rejecting HM2. The results also revealed that PPV significantly mediated the relationship between PUS and IWM (β = 0.128, p = 0.000), supporting the acceptance of HM3. Similarly, PPV mediated the relationship between PTA and IWM (β = 0.166, p = 0.000), thus, supporting HM4. Table 4 presents all the mediation analysis results.

## 5. Discussion

The present study aimed to identify the personal health behaviour and technological factors that influenced Chinese adults' intention to use and adopt WMD. Findings confirmed that PCT, PUS, and PTA significantly affected the PPV of the WMDs. This study's results coincide with Asadi et al. [22] who reports that technology compatibility and usefulness are significant predictors that build the PPV of wearable health technologies. Furthermore, the PTA also

**Table 4. Mediating effects.**

| Hyp. | Path | Coefficients | CI–Min | CI–Max | t | P | Decision |
|------|------|-------------|--------|--------|---|---|----------|
| HM1 | PCT → PPV → IWM | 0.154 | 0.125 | 0.183 | 8.740 | 0.000 | Mediation |
| HM2 | PCO → PPV → IWM | 0.012 | -0.008 | 0.033 | 0.979 | 0.164 | No Mediation |
| HM3 | PUS → PPV → IWM | 0.128 | 0.101 | 0.158 | 7.326 | 0.000 | Mediation |
| HM4 | PTA → PPV → IWM | 0.166 | 0.135 | 0.199 | 8.579 | 0.000 | Mediation |

**Note**: HCS: Health consciousness; HAY: Health anxiety; PCT: Perceived compatibility; PCO: Perceived cost; PUS: Perceived usefulness; PTA: Perceived technology accuracy; PPV: Perceived product value; PCM: Perceived critical mass; IWM: Intention to Use WMD; AWM: adoption of WMD.

**Source**: Author's data analysis.

affected the PPV. This finding concurs with Lee and Lee's [1] work that accurately performing technologies enhance the perception of value of novel healthcare technologies. Nonetheless, the effect of PCO on PPV was not statistically significant. The respondents perceived the WMDs as costly and this had reduced the PPV among prospective users. This current study's finding matches the result in Bandara and Amarasens [23], whereby the perception of higher cost reduces the PPV.

The present study's finding agrees with the outcome in Naami et al. [31] whereby the technology's PPV influences the intention to use portable technological devices. Additionally, HCS significantly instigated the intention to use the WMDs. This outcome concurs with the result in a study by Sergueeva et al. [3] who reports that personal HCS impacts the formation of the intention to use WMDs. Furthermore, the mass adoption of technology facilitates the other users to build the intention to use the technology [30]. The current investigation's result confirmed that PCM significantly facilitated the formation of the intention to use the WMDs. The finding matches with the result in Yen et al. [14] which reports that the social acceptability of the technology influences the new users' technology adoption. This suggests that the public's mass acceptance depicts the sense of confidence and assurance that the technology is good to use. Nevertheless, this current study's result revealed that HAY insignificantly influenced the intention to use the WMDs. This finding disagrees with the result of Fanbo et al. [37] who report that HAY might not lead to the formation of the intention to use the WMDs.

Lastly, the analysis confirmed that PCM and IWM significantly influenced the adoption of WMDs. The findings agree with the outcomes in Gong et al. [16] whereby technology adoption by family and friends provides the trust and confidence that the adoption of the technology can benefit the users and is necessary to reduce the burden on healthcare institutions. Moreover, the intention to use the WMDs significantly influenced the adoption of WMDs among the study samples. This outcome concurs with the result in Dehghani et al. [17] which reports that behavioural intention is a strong predictor of technology adoption behaviour.

On the other hand, the mediation analysis confirmed that the PPV insignificantly mediated the relationship between PCO and IWM. Nonetheless, the PPV mediated the relationships between PCT and IWM, PUS and IWM, and PTA and IWM. The findings confirmed that the WMDs' attributes facilitated the intention to use the WMDs through the PPV. The perception of value influenced the behavioural intention to use the WMDs.

## 6. Conclusion

The current study explored the value perception built for healthcare technologies with WMDs' attributes, health behaviour, healthcare technology attributes, and mass adoption

that prompted the intention and adoption of WMDs. This study's findings offer valuable insights into how senior adult consumers adopt WMDs. The present study's prevalent theoretical and managerial contributions along with the limitations of the study are discussed as follows.

### Theoretical implications

This study extends the TAM to perceive that the technology's value emerges from the technological attributes. The perception of value builds the understanding that the technology is valuable and can influence the intention to use the technology. Furthermore, the perception of value creates the understanding for the acquisition, transaction, and consumption of the technology that is a subjective evaluation of the benefits that a consumer may gain from the healthcare product or services [31].

Moreover, this study establishes that the critical mass adoption as a social influence affects the intention and later the adoption of healthcare technology. Mass adoption builds the social acceptance of technology and promotes the intention to use and the adoption of healthcare technologies. Healthcare technologies are crucial for personal health responsibility and can reduce the burden on the public healthcare system. The general public can take an active part in limiting the need for resources to deliver quality healthcare services to the global population.

### Practical and managerial implications

This study has various managerial implications for marketers, designers, and developers of WMDs that may help them meet users' needs and wants. First, the firms need to design and develop the WMDs with reduced cost as the perception of cost significantly lowers the product value and later the adoption of WMDs. Second, they need to promote WMDs to critically ill and elderly general users increasing overall health consciousness and confidence to use the WMDs. The mass marketing drive helps reach the divergent segments of prospective users and develops the proper awareness that promotes the intention to use and later the adoption of WMDs. To achieve this, mass adoption by the general public through advertisement and marketing campaigns is the right choice. Mass adoption not only facilitates the adoption of WMDs but also reduces the burden on the public healthcare system, especially during major health issues such as COVID-19.

### Study limitations

This study has three limitations. First, the current study employed only limited factors that influenced the value perception of WMDs. Thus, more pertinent factors need to be examined to explore WMDs' value development. Second, the current study explored the intention to use and the adoption of WMDs. However, since technology adoption changes over time, future studies need to explore the continuous intention to use the WMDs among the respondents. Lastly, the current study employed a quantitative design, offering limited exposure to the phenomenon examined. Therefore, future works may utilise the mixed-method research design to fully explore and form a broader awareness of healthcare technology adoption. Furthermore, it would be interesting to discover the impact of personal health behaviour on the adoption of healthcare technologies to facilitate the public healthcare system.

**Appendix 1. Survey instrument.**

| | | |
|---|---|---|
| *Health Consciousness (HCS)* | | |
| HCS1 | I think my health depends on how well I take care of myself. | [49,50] |
| HCS2 | I am actively engaged in the prevention of disease and illness. | |
| HCS3 | I think taking preventive measures help to stay healthy. | |
| HCS4 | Living a healthy life is important to me. | |
| HCS5 | I am constantly examining my health. | |
| *Health Anxiety (HAY)* | | |
| HAY1 | I usually anxious about my health. | [51] |
| HAY2 | I am worried about my health condition. | |
| HAY3 | Thinking about my health leaves me with uneasy feelings. | |
| HAY4 | I frequently worry about my health. | |
| HAY5 | I feel concerned whenever I reflect on the status of my physical health. | |
| *Perceived Compatibility (PCT)* | | |
| PCT1 | Using wearable medical devices would be compatible with my lifestyle. | [52] |
| PCT2 | I think that using wearable medical devices would fit well with the way I work and live. | |
| PCT3 | I think using wearable medical devices suits my way of managing health at home. | |
| PCT4 | I think the wearable medical device is very much compatible with my lifestyle. | |
| *Perceived Cost (PCO)* | | |
| PCO1 | Wearable medical devices are not cheap. | [52] |
| PCO2 | Wearable medical devices are unreasonably priced. | |
| PCO3 | I am not satisfied with the price that I paid for the wearable medical device. | |
| *Perceived Usefulness (PUS)* | | |
| PUS1 | Using wearable medical devices enables me to check my health condition quickly. | [12] |
| PUS2 | Using wearable medical devices makes it easier to accomplish my health condition checking. | |
| PUS3 | Using wearable medical device save my time and effort. | |
| *Perceived Technology Accuracy (PTA)* | | |
| PTA1 | I can rely on the health services provided by wearable medical devices. | [13,53] |
| PTA2 | I am wearable medical devices offers consistent results over time. | |
| PTA3 | I think wearable medical devices have good working standards continuously. | |
| PTA4 | I think wearable medical devices are reliable. | |
| PTA5 | I feel confident that wearable medical devices are offering error-free results. | |
| *Perceived Product Value (PPV)* | | |
| PPV1 | Wearable medical devices are beneficial | [52] |
| PPV2 | Using wearable medical devices valuable to me. | |
| PPV3 | I think the wearable medical device is worthwhile. | |
| PPV4 | Overall, using wearable medical devices delivers good value to me. | |
| *Perceived Critical Mass (PCM)* | | |
| PCM1 | Most people in my group use wearable medical devices. | [12] |
| PCM2 | Many people to whom I usually communicate are using wearable medical devices. | |
| PCM3 | Most people in my community are using wearable medical devices frequently. | |
| PCM4 | I know many people having health issues are using wearable medical devices frequently. | |
| *Intention to Use WMDs (IWM)* | | |
| IWM1 | I intend to use wearable medical devices to manage my health in the future. | [4,13,19] |
| IWM2 | I will always try to use wearable medical devices to manage my health in my daily life in the future. | |
| IWM3 | I plan to use wearable medical devices frequently to manage my health in the future. | |
| IWM4 | I would be willing to develop a habit to use wearable medical devices soon. | |
| IWM5 | I predict I will use wearable medical devices to manage my health information. | |

*(Continued)*

**Appendix 1.** (Continued)

| *Adoption of WMD (AWM)* | | |
|---|---|---|
| AWM | I am actively using wearable medical devices. | [54] |

**Note**: HCS: Health consciousness; HAY: Health anxiety; PCT: Perceived compatibility; PCO: Perceived cost; PUS: Perceived usefulness; PTA: Perceived technology accuracy; PPV: Perceived product value; PCM: Perceived critical mass; IWM: Intention to Use WMD; AWM: adoption of WMD.

**Appendix 2. Discriminant validity.**

| | HCS | HAY | PCT | PCO | PUS | PTA | PPV | PCM | IWM | AWM |
|---|---|---|---|---|---|---|---|---|---|---|
| Fornell-Larcker Criterion | | | | | | | | | | |
| HCS | 0.734 | | | | | | | | | |
| HAY | 0.505 | 0.768 | | | | | | | | |
| PCT | 0.558 | 0.557 | 0.759 | | | | | | | |
| PCO | 0.242 | 0.359 | 0.486 | 0.891 | | | | | | |
| PUS | 0.560 | 0.445 | 0.569 | 0.339 | 0.775 | | | | | |
| PTA | 0.464 | 0.409 | 0.546 | 0.403 | 0.562 | 0.707 | | | | |
| PPV | 0.545 | 0.461 | 0.654 | 0.399 | 0.663 | 0.661 | 0.761 | | | |
| PCM | 0.439 | 0.506 | 0.680 | 0.502 | 0.490 | 0.511 | 0.584 | 0.736 | | |
| IWM | 0.485 | 0.428 | 0.608 | 0.387 | 0.577 | 0.577 | 0.699 | 0.581 | 0.761 | |
| AWM | 0.313 | 0.315 | 0.522 | 0.346 | 0.443 | 0.460 | 0.559 | 0.495 | 0.603 | 1.000 |
| HTMT Ratio | | | | | | | | | | |
| HCS | - | | | | | | | | | |
| HAY | 0.625 | - | | | | | | | | |
| PCT | 0.721 | 0.706 | - | | | | | | | |
| PCO | 0.314 | 0.460 | 0.650 | - | | | | | | |
| PUS | 0.773 | 0.598 | 0.805 | 0.483 | - | | | | | |
| PTA | 0.601 | 0.521 | 0.727 | 0.543 | 0.797 | - | | | | |
| PPV | 0.705 | 0.584 | 0.868 | 0.535 | 0.896 | 0.874 | - | | | |
| PCM | 0.571 | 0.658 | 0.900 | 0.691 | 0.704 | 0.695 | 0.791 | - | | |
| IWM | 0.601 | 0.519 | 0.772 | 0.496 | 0.784 | 0.735 | 0.890 | 0.753 | - | |
| AWM | 0.351 | 0.346 | 0.600 | 0.402 | 0.543 | 0.531 | 0.646 | 0.584 | 0.666 | - |
| Loading and Cross-Loading | | | | | | | | | | |
| HCS1 | *0.756* | 0.385 | 0.395 | 0.171 | 0.386 | 0.356 | 0.365 | 0.336 | 0.340 | 0.206 |
| HCS2 | *0.706* | 0.341 | 0.430 | 0.205 | 0.417 | 0.294 | 0.418 | 0.325 | 0.344 | 0.263 |
| HCS3 | *0.782* | 0.359 | 0.367 | 0.140 | 0.436 | 0.351 | 0.388 | 0.274 | 0.357 | 0.201 |
| HCS4 | *0.692* | 0.394 | 0.399 | 0.138 | 0.394 | 0.315 | 0.363 | 0.275 | 0.319 | 0.207 |
| HCS5 | *0.731* | 0.376 | 0.451 | 0.226 | 0.418 | 0.378 | 0.452 | 0.389 | 0.410 | 0.266 |
| HAY1 | 0.416 | *0.781* | 0.425 | 0.273 | 0.363 | 0.313 | 0.364 | 0.360 | 0.319 | 0.243 |
| HAY2 | 0.464 | *0.757* | 0.419 | 0.252 | 0.375 | 0.331 | 0.358 | 0.363 | 0.347 | 0.247 |
| HAY3 | 0.321 | *0.741* | 0.430 | 0.302 | 0.279 | 0.304 | 0.326 | 0.412 | 0.317 | 0.214 |
| HAY4 | 0.367 | *0.767* | 0.440 | 0.269 | 0.350 | 0.299 | 0.350 | 0.389 | 0.330 | 0.244 |
| HAY5 | 0.366 | *0.793* | 0.425 | 0.284 | 0.336 | 0.323 | 0.371 | 0.420 | 0.326 | 0.260 |
| PCT1 | 0.401 | 0.404 | *0.745* | 0.379 | 0.405 | 0.374 | 0.481 | 0.507 | 0.455 | 0.358 |
| PCT2 | 0.403 | 0.461 | *0.776* | 0.356 | 0.413 | 0.442 | 0.482 | 0.521 | 0.443 | 0.408 |
| PCT3 | 0.463 | 0.408 | *0.751* | 0.350 | 0.412 | 0.416 | 0.508 | 0.497 | 0.453 | 0.376 |
| PCT4 | 0.426 | 0.421 | *0.765* | 0.390 | 0.493 | 0.425 | 0.513 | 0.539 | 0.493 | 0.443 |
| PCO1 | 0.229 | 0.325 | 0.437 | *0.891* | 0.305 | 0.360 | 0.355 | 0.473 | 0.337 | 0.292 |

*(Continued)*

**Appendix 2.** (Continued)

| | | | | | | | | | | |
|---|---|---|---|---|---|---|---|---|---|---|
| PCO2 | 0.203 | 0.315 | 0.429 | *0.809* | 0.299 | 0.358 | 0.356 | 0.422 | 0.352 | 0.325 |
| PCO3 | 0.222 | 0.352 | 0.259 | *0.808* | 0.305 | 0.538 | 0.656 | 0.242 | 0.523 | 0.449 |
| PUS1 | 0.406 | 0.365 | 0.438 | 0.265 | *0.824* | 0.441 | 0.479 | 0.392 | 0.458 | 0.365 |
| PUS2 | 0.461 | 0.364 | 0.424 | 0.273 | *0.810* | 0.429 | 0.522 | 0.398 | 0.439 | 0.351 |
| PUS3 | 0.431 | 0.302 | 0.461 | 0.249 | *0.684* | 0.437 | 0.466 | 0.346 | 0.444 | 0.311 |
| PTA1 | 0.367 | 0.305 | 0.388 | 0.271 | 0.394 | *0.719* | 0.446 | 0.366 | 0.404 | 0.307 |
| PTA2 | 0.289 | 0.285 | 0.346 | 0.285 | 0.397 | *0.707* | 0.488 | 0.354 | 0.419 | 0.312 |
| PTA3 | 0.325 | 0.253 | 0.403 | 0.319 | 0.380 | *0.678* | 0.436 | 0.365 | 0.388 | 0.324 |
| PTA4 | 0.325 | 0.272 | 0.379 | 0.248 | 0.403 | *0.716* | 0.530 | 0.355 | 0.403 | 0.322 |
| PTA5 | 0.338 | 0.339 | 0.421 | 0.310 | 0.412 | *0.714* | 0.421 | 0.370 | 0.426 | 0.364 |
| PPV1 | 0.406 | 0.372 | 0.500 | 0.320 | 0.482 | 0.518 | *0.825* | 0.464 | 0.534 | 0.418 |
| PPV2 | 0.430 | 0.367 | 0.493 | 0.281 | 0.490 | 0.495 | *0.784* | 0.438 | 0.537 | 0.400 |
| PPV3 | 0.412 | 0.318 | 0.499 | 0.290 | 0.446 | 0.496 | *0.646* | 0.422 | 0.487 | 0.430 |
| PPV4 | 0.407 | 0.342 | 0.495 | 0.320 | 0.504 | 0.502 | *0.777* | 0.449 | 0.562 | 0.451 |
| PCM1 | 0.304 | 0.377 | 0.488 | 0.374 | 0.347 | 0.354 | 0.409 | *0.765* | 0.404 | 0.365 |
| PCM2 | 0.226 | 0.379 | 0.475 | 0.409 | 0.298 | 0.352 | 0.387 | *0.703* | 0.366 | 0.348 |
| PCM3 | 0.304 | 0.375 | 0.517 | 0.325 | 0.357 | 0.385 | 0.439 | *0.738* | 0.433 | 0.387 |
| PCM4 | 0.439 | 0.359 | 0.517 | 0.376 | 0.427 | 0.408 | 0.474 | *0.736* | 0.494 | 0.357 |
| IWM1 | 0.384 | 0.321 | 0.484 | 0.294 | 0.483 | 0.447 | 0.562 | 0.464 | *0.775* | 0.443 |
| IWM2 | 0.381 | 0.330 | 0.462 | 0.312 | 0.444 | 0.414 | 0.505 | 0.441 | *0.749* | 0.473 |
| IWM3 | 0.344 | 0.314 | 0.454 | 0.262 | 0.405 | 0.401 | 0.521 | 0.412 | *0.765* | 0.481 |
| IWM4 | 0.373 | 0.343 | 0.450 | 0.314 | 0.450 | 0.485 | 0.535 | 0.431 | *0.726* | 0.484 |
| IWM5 | 0.362 | 0.318 | 0.459 | 0.286 | 0.408 | 0.443 | 0.532 | 0.459 | *0.787* | 0.409 |
| AWM | 0.313 | 0.315 | 0.522 | 0.346 | 0.443 | 0.460 | 0.559 | 0.495 | 0.603 | *1.000* |

**Note**: HCS: Health consciousness; HAY: Health anxiety; PCT: Perceived compatibility; PCO: Perceived cost; PUS: Perceived usefulness; PTA: Perceived technology accuracy; PPV: Perceived product value; PCM: Perceived critical mass; IWM: Intention to Use WMD; AWM: adoption of WMD.

# Supporting information

**S1 Data.**
(CSV)

# Author Contributions

**Conceptualization:** Qing Yang, Abdullah Al Mamun, Naeem Hayat, Mohd Fairuz Md. Salleh, Gao Jingzu, Noor Raihani Zainol.

**Data curation:** Qing Yang, Mohd Fairuz Md. Salleh.

**Formal analysis:** Abdullah Al Mamun, Naeem Hayat, Gao Jingzu, Noor Raihani Zainol.

**Methodology:** Qing Yang, Mohd Fairuz Md. Salleh, Noor Raihani Zainol.

**Writing – original draft:** Qing Yang, Gao Jingzu, Noor Raihani Zainol.

**Writing – review & editing:** Abdullah Al Mamun, Naeem Hayat, Mohd Fairuz Md. Salleh.

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
