## [Decision Letter · Decision Letter 0]

25 Apr 2022

PONE-D-22-07166Modelling the Mass Adoption Potential of Wearable Medical DevicesPLOS ONE

Thank you for submitting your manuscript to PLOS ONE. After careful consideration, we feel that it has merit but does not fully meet PLOS ONE’s publication criteria as it currently stands. Therefore, we invite you to submit a revised version of the manuscript that addresses the points raised during the review process.

We look forward to receiving your revised manuscript.

Kind regards,

Luigi Lavorgna

Academic Editor

PLOS ONE

Journal Requirements:

When submitting your revision, we need you to address these additional requirements. 1. Please ensure that your manuscript meets PLOS ONE's style requirements, including those for file naming. The PLOS ONE style templates can be found at https://journals.plos.org/plosone/s/file?id=wjVg/PLOSOne_formatting_sample_main_body.pdf and https://journals.plos.org/plosone/s/file?id=ba62/PLOSOne_formatting_sample_title_authors_affiliations.pdf 2. We suggest you thoroughly copyedit your manuscript for language usage, spelling, and grammar. If you do not know anyone who can help you do this, you may wish to consider employing a professional scientific editing service.  Whilst you may use any professional scientific editing service of your choice, PLOS has partnered with both American Journal Experts (AJE) and Editage to provide discounted services to PLOS authors. Both organizations have experience helping authors meet PLOS guidelines and can provide language editing, translation, manuscript formatting, and figure formatting to ensure your manuscript meets our submission guidelines. To take advantage of our partnership with AJE, visit the AJE website (http://learn.aje.com/plos/) for a 15% discount off AJE services. To take advantage of our partnership with Editage, visit the Editage website (www.editage.com) and enter referral code PLOSEDIT for a 15% discount off Editage services.  If the PLOS editorial team finds any language issues in text that either AJE or Editage has edited, the service provider will re-edit the text for free. Upon resubmission, please provide the following: The name of the colleague or the details of the professional service that edited your manuscript A copy of your manuscript showing your changes by either highlighting them or using track changes (uploaded as a *supporting information* file) A clean copy of the edited manuscript (uploaded as the new *manuscript* file) 3. Please include in the Methods section of your manuscript text the information you have provided in the Ethics Statement section of the submission form regarding 1) the assessment by your institutional ethics committee, and 2) the consent obtained from participants. Please also include as a supporting information file the questionnaire used in the online survey. 4. We note that you have stated that you will provide repository information for your data at acceptance. Should your manuscript be accepted for publication, we will hold it until you provide the relevant accession numbers or DOIs necessary to access your data. If you wish to make changes to your Data Availability statement, please describe these changes in your cover letter and we will update your Data Availability statement to reflect the information you provide. 5. Please amend your authorship list in your manuscript file to include authors Abdullah Al Mamun, 
Yang Qing, Naeem Hayat, Mohd Fairuz Md. Salkeh, Gao Jingzu and Noor Raihani Zaino. 6. Please include your full ethics statement in the ‘Methods’ section of your manuscript file. In your statement, please include the full name of the IRB or ethics committee who approved or waived your study, as well as whether or not you obtained informed written or verbal consent. If consent was waived for your study, please include this information in your statement as well.  7. Please include captions for your Supporting Information files at the end of your manuscript, and update any in-text citations to match accordingly. Please see our Supporting Information guidelines for more information: http://journals.plos.org/plosone/s/supporting-information. 

Reviewers' comments:

Reviewer's Responses to Questions

**Comments to the Author**

1. Is the manuscript technically sound, and do the data support the conclusions?

Reviewer #1: Yes

2. Has the statistical analysis been performed appropriately and rigorously? 

Reviewer #1: Yes

3. Have the authors made all data underlying the findings in their manuscript fully available?

Reviewer #1: Yes

4. Is the manuscript presented in an intelligible fashion and written in standard English?

Reviewer #1: No

5. Review Comments to the Author

Reviewer #1: 1) “The public requires promising health facilities as public health services are may not be available for all when they needed.” The sentence is not phrased correctly, rewrite it as follows: The public requires promising health facilities as public health services may not be available to everyone when they might need them.

2) “However, the question remains how the WMD technological attributes, personal health behaviour, and social mass adoption instigate the intention to use and adopt WMDs.” Rephrase as: However, the question remains how the WMD technological attributes, personal health behaviour, and social mass adoption instigate the intention to use and adopt WMDs.

3) “Besides, this study examines the development of the intention to use with personal health behaviour [including health consciousness (HCS) and health anxiety (HAY)], perceived product value (PPV), and mass adoption of WMDs.” Please try to be clearer in explaining the purpose of the study.

4) “The mass adoption among the community and peers builds the perception of acceptance and consideration that the technology is good and benefits the users.” Please try to use less generic and more scientific terms.

5) “The degree to which a person believes that using WMDs would cost money is characterised as PCO.” Perhaps it would be more appropriate to use "defined as" instead of "characterised".

6) “Besides, the health advantages that a user expects from a digital health device are referred to as PUS.” It seems a bit redundant like many other sentences in the text. There is no point in rewriting the previous sentence by changing the word order.

7) “Buyers and producers have progressively acknowledged the significance of perceived value in behaviour intention since it is considered one of the most important factors in behaviour intention.” Would it be better to phrase it this way? “Perceived value is considered one of the most important factors in behavioural intention, to the extent that buyers and producers have gradually recognised its great importance.”

8) “H2c: HAY has a positive effect on the intention to use WMDs.

2.2.8 Perceived critical mass (PCM)”. Please insert a space between these two lines, as for the previous paragraphs.

9) “The level of a user’s apparent desire to use new technology is referred to as behavioural intention.” It would be better to include the definition of "behavior intention" in paragraph 2.2.5 when you first mention it.

10) Jiangsu 1100 8.6 The value in the table is incorrect.

11) “This study’s findings offer valuable insights into how

elderly consumers adopt WMDs.” Only 6.5% of the respondents were over 60 years old.

12) In the introduction section, the authors should mention more extensively the numerous experiences available concerning the use of WD in the management of health (suggested ref: PMID: 34018047; PMID: 30405913; PMID: 33802029).

6. PLOS authors have the option to publish the peer review history of their article (what does this mean?). If published, this will include your full peer review and any attached files.

Reviewer #1: No

---

## [Author Response · Author response to Decision Letter 0]

4 May 2022

Reply to Reviewer(s) Comments

Modelling the Mass Adoption Potential of Wearable Medical Devices

PONE-D-22-07166

 Journal Requirements:

1. Please ensure that your manuscript meets PLOS ONE's style requirements, including those for file naming. The PLOS ONE style templates can be found at:

Author(s) Reply: I will do it.

Author(s) Reply:

We uploaded proof of editing and certificate from the editor as *supporting information*

3. Please include in the Methods section of your manuscript text the information you have provided in the Ethics Statement section of the submission form regarding 1) the assessment by your institutional ethics committee, and 2) the consent obtained from participants.

Please also include as a supporting information file the questionnaire used in the online survey.

Author(s) Reply: Amended Accordingly. Complete questionnaire presented as Appendix 1 (before references)

Amendment in the Manuscript: 

3. RESEARCH METHODOLOGY

The local ethics committee (Universiti Malaysia Kelantan) ruled that no formal ethics approval was required in this particular case because this research did not collect any medical information, there was no known risk involved, no intention to publish personal information, and did not collect data from underaged respondents. This study has been performed in accordance with the Declaration of Helsinki. Written informed consent for participation was obtained from respondents who participated in the survey. For the respondents who participated in the survey online (using Google form), they were asked to read the ethical statement posted at the top of the form (There is no compensation for responding, nor is there any known risk. In order to ensure that all information will remain confidential, please do not include your name. Participation is strictly voluntary and you may refuse to participate at any time. and proceed only if they agree. No data was collected from anyone under 18 years old.

Author(s) Reply: We will submit the questionnaire and data as supporting material. Please update the Data Availability Statement accordingly.

5. Please amend your authorship list in your manuscript file to include authors Abdullah Al Mamun, Yang Qing, Naeem Hayat, Mohd Fairuz Md. Salkeh, Gao Jingzu and Noor Raihani Zaino.

Author(s) Reply: Amended Accordingly.

Amendment in the Manuscript: 

Qing Yang1, Abdullah Al Mamun1*, Naeem Hayat2, Mohd Fairuz Md. Salleh1, Gao Jingzu3 and Noor Raihani Zainol4

1UKM - Graduate School of Business, Universiti Kebangsaan Malaysia, 43600, UKM Bangi, Malaysia; 2Global Entrepreneurship Research and Innovation Centre, Universiti Malaysia Kelantan, 16100 Kota Bharu, Malaysia; 3UCSI Graduate Business School, UCSI University, Cheras, 56000 Kuala Lumpur, Malaysia; 4Faculty of Entrepreneurship and Business, Universiti Malaysia Kelantan, 16100 Kota Bharu, Malaysia

* almamun@ukm.edu.my, mamun7793@gmail.com

Author(s) Reply: Amended Accordingly.

Amendment in the Manuscript: 

3. RESEARCH METHODOLOGY

The local ethics committee (Universiti Malaysia Kelantan) ruled that no formal ethics approval was required in this particular case because this research did not collect any medical information, there was no known risk involved, no intention to publish personal information, and did not collect data from underaged respondents. This study has been performed in accordance with the Declaration of Helsinki. Written informed consent for participation was obtained from respondents who participated in the survey. For the respondents who participated in the survey online (using Google form), they were asked to read the ethical statement posted at the top of the form (There is no compensation for responding, nor is there any known risk. In order to ensure that all information will remain confidential, please do not include your name. Participation is strictly voluntary and you may refuse to participate at any time. and proceed only if they agree. No data was collected from anyone under 18 years old.

Author(s) Reply: Amended Accordingly.

Amendment in the Manuscript: The raw data supporting the conclusions of this article is presented as supporting material: S1 Data – WMD.

REVIEWERS' COMMENTS:

Comments to the Author

1. Is the manuscript technically sound, and do the data support the conclusions?

Reviewer #1: Yes

2. Has the statistical analysis been performed appropriately and rigorously? 

Reviewer #1: Yes

3. Have the authors made all data underlying the findings in their manuscript fully available?

Reviewer #1: Yes

4. Is the manuscript presented in an intelligible fashion and written in standard English?

Reviewer #1: No

5. Review Comments to the Author

Comment 1: “The public requires promising health facilities as public health services are may not be available for all when they needed.” The sentence is not phrased correctly, rewrite it as follows: The public requires promising health facilities as public health services may not be available to everyone when they might need them.

Author(s) Reply: Amended Accordingly. Thank you very much for your thoughtful suggestions.

Comment 2: “However, the question remains how the WMD technological attributes, personal health behaviour, and social mass adoption instigate the intention to use and adopt WMDs.” Rephrase as: However, the question remains how the WMD technological attributes, personal health behaviour, and social mass adoption instigate the intention to use and adopt WMDs.

Author(s) Reply: Amended Accordingly. Thank you very much for your thoughtful suggestions.

Comment 3: “Besides, this study examines the development of the intention to use with personal health behaviour [including health consciousness (HCS) and health anxiety (HAY)], perceived product value (PPV), and mass adoption of WMDs.” Please try to be clearer in explaining the purpose of the study.

Author(s) Reply: Amended Accordingly. Thank you very much for your thoughtful suggestions.

Amendment in the Manuscript:

However, the question remains how the WMD technological attributes, personal health behaviour, and social mass adoption instigate the intention to use and adopt WMDs. Therefore, the current study explores the construction of WMD value with the technology features (compatibility, cost, usefulness, and accuracy) forming the product value. Besides, this study examines the effects of health consciousness, health anxiety, perceived product value, and perceived critical mass on the intention and adoption of WMDs.

Comment 4: “The mass adoption among the community and peers builds the perception of acceptance and consideration that the technology is good and benefits the users.” Please try to use less generic and more scientific terms.

Author(s) Reply: Amended Accordingly. Thank you very much for your thoughtful suggestions.

Amendment in the Manuscript:

The perception of acceptance and consideration that the technology is useful and benefits the users grows as a result of widespread adoption among the community and peers.

Comment 5: “The degree to which a person believes that using WMDs would cost money is characterised as PCO.” Perhaps it would be more appropriate to use "defined as" instead of "characterised".

Author(s) Reply: Amended Accordingly. Thank you very much for your thoughtful suggestions.

Amendment in the Manuscript:

The degree to which a person believes that using WMDs would cost money is defined as PCO (Liu, Wang, Huang, Huang, Yang, & Li, 2019). The higher the perceived cost of WMDs, the less likely they will be used.

Comment 6: “Besides, the health advantages that a user expects from a digital health device are referred to as PUS.” It seems a bit redundant like many other sentences in the text. There is no point in rewriting the previous sentence by changing the word order.

Author(s) Reply: Amended Accordingly. Thank you very much for your thoughtful suggestions. We removed this part.

Comment 7: “Buyers and producers have progressively acknowledged the significance of perceived value in behaviour intention since it is considered one of the most important factors in behaviour intention.” Would it be better to phrase it this way? “Perceived value is considered one of the most important factors in behavioural intention, to the extent that buyers and producers have gradually recognised its great importance.”

Author(s) Reply: Amended Accordingly. Thank you very much for your thoughtful suggestions. 

Comment 8: “H2c: HAY has a positive effect on the intention to use WMDs.

2.2.8 Perceived critical mass (PCM)”. Please insert a space between these two lines, as for the previous paragraphs.

Author(s) Reply: Amended Accordingly. Thank you very much for your thoughtful suggestions. 

Comment 9: “The level of a user’s apparent desire to use new technology is referred to as behavioural intention.” It would be better to include the definition of "behavior intention" in paragraph 2.2.5 when you first mention it.

Author(s) Reply: Amended Accordingly. Thank you very much for your thoughtful suggestions. 

Amendment in the Manuscript:

In contrast, "behavioural intention" refers to a user's apparent desire to adopt new technology (Chang, 2020; Venkatesh, Thing, & Xu, 2012). Perceived product value is considered one of the most important factors in behavioural intention, to the extent that buyers and producers have gradually recognised its great importance (Naami et al., 2017). According to numerous information technology studies, the perceived value of using mobile internet services on portable devices has a beneficial impact on adoption intention (Yang et al., 2016; Naami et al., 2017).

Comment 10: Jiangsu 1100 8.6 The value in the table is incorrect.

Author(s) Reply: Amended Accordingly. Thank you very much for your thoughtful suggestions. 

Amendment in the Manuscript:

100 (8.6)

Comment 11: “This study’s findings offer valuable insights into how

elderly consumers adopt WMDs.” Only 6.5% of the respondents were over 60 years old.

Author(s) Reply: Amended Accordingly. Thank you very much for your thoughtful suggestions. 

Amendment in the Manuscript:

Changed to senior adult as around 85% of the respondents are more than 40 years old.

Comment 12: In the introduction section, the authors should mention more extensively the numerous experiences available concerning the use of WD in the management of health (suggested ref: PMID: 34018047; PMID: 30405913; PMID: 33802029).

Author(s) Reply: Amended Accordingly. Thank you very much for your thoughtful suggestions. 

Amendment in the Manuscript:

Currently available smartphone-based WMDs include various types of motion sensors and bio-sensors that can record feedback on a patient's mobility status and other physical activities based on data collected on a daily basis, allowing for a more objective assessment of possible therapies (Abbadessa et al., 2021; Sparaco, 2018). The WMD's data can be utilized alone or in combination with pharmaceuticals, devices, or other treatments to improve patient care and health outcomes, especially in the case of chronic conditions (Abbadessa, 2022).

Reference:

Abbadessa, G., Brigo, F., Clerico, M., De Mercanti, S., Trojsi, F., Tedeschi, G., … Lavorgna, L. (2022). Digital therapeutics in neurology. Journal of Neurology, 269(3), 1209-1224. doi:10.1007/s00415-021-10608-4

Abbadessa, G., Lavorgna, L., Miele, G., Mignone, A., Signoriello, E., Lus, G., … Bonavita, S. (2021). Assessment of multiple sclerosis disability progression using a wearable biosensor: A pilot study. Journal of Clinical Medicine, 10(6), 1160. doi:10.3390/jcm10061160

Sparaco, M., Lavorgna, L., Conforti, R., Tedeschi, G., & Bonavita, S. (2018). The role of wearable devices in multiple sclerosis. Multiple Sclerosis International, 2018, 1-7. doi:10.1155/2018/7627643

---

## [Editor Report · Decision Letter 1]

18 May 2022

Modelling the Mass Adoption Potential of Wearable Medical Devices

PONE-D-22-07166R1

We’re pleased to inform you that your manuscript has been judged scientifically suitable for publication and will be formally accepted for publication once it meets all outstanding technical requirements.

Kind regards,

Luigi Lavorgna

Academic Editor

PLOS ONE
---

## [Editor Report · Acceptance letter]

23 May 2022

PONE-D-22-07166R1 

Modelling the Mass Adoption Potential of Wearable Medical Devices 

Dear Dr. Al Mamun:

I'm pleased to inform you that your manuscript has been deemed suitable for publication in PLOS ONE. Congratulations! Your manuscript is now with our production department. 

Kind regards, 

on behalf of

Dr. Luigi Lavorgna 

Academic Editor

PLOS ONE